# Effects of Extended Viewing Distance on Accommodative Response and Pupil Size of Myopic Adults by Using a Double-Mirror System

**DOI:** 10.3390/ijerph19052942

**Published:** 2022-03-03

**Authors:** Shu-Yuan Lin, Hui-Rong Su, Chen-Cheng Lo, Shang-Min Yeh, Chi-Hung Lee, Richard Wu, Fen-Chi Lin, Yen-Wei Chu, Shuan-Yu Huang

**Affiliations:** 1Ph.D. Program in Medical Biotechnology, National Chung Hsing University, Taichung 402, Taiwan; d106001762@mail.nchu.edu.tw; 2Department of Optometry, Chung Shan Medical University, Taichung 402, Taiwan; leah860307@gmail.com (H.-R.S.); s0885013@gm.csmu.edu.tw (C.-C.L.); ysm@csmu.edu.tw (S.-M.Y.); 3Department of Ophthalmology, Chung Shan Medical University Hospital, Taichung 402, Taiwan; 4Department of Electrical Engineering, Feng Chia University, Taichung 407, Taiwan; chihlee@fcu.edu.tw; 5College of Optometry and Visual Science, Pacific University, Forest Grove, OR 97126, USA; richard.wu@brightenoptix.com; 6Brighten Optix Research and Development Center, Taipei 111, Taiwan; 7Department of Ophthalmology, Kaohsiung Armed Forced General Hospital, Kaohsiung 802, Taiwan; a0898020012@mail.802.org.tw; 8Institute of Genomics and Bioinformatics, National Chung Hsing University, Taichung 402, Taiwan; 9Institute of Molecular Biology, National Chung Hsing University, Taichung 402, Taiwan; 10Agricultural Biotechnology Center, National Chung Hsing University, Taichung 402, Taiwan; 11Biotechnology Center, National Chung Hsing University, Taichung 402, Taiwan; 12Ph.D. Program in Translational Medicine, National Chung Hsing University, Taichung 402, Taiwan; 13Rong Hsing Research Center for Translational Medicine, National Chung Hsing University, Taichung 402, Taiwan

**Keywords:** accommodative response, pupil size, double-mirror system (DMS), accommodative microfluctuations (AMFs), fatigue, near work, myopia

## Abstract

Purposes: This study discussed the accommodative response and pupil size of myopic adults using a double-mirror system (DMS). The viewing distance could be extended to 2.285 m by using a DMS, which resulted in a reduction and increase in the accommodative response and pupil size, respectively. By using a DMS, the reduction of the accommodative response could improve eye fatigue with near work. Method: Sixty subjects aged between 18 and 22 years old were recruited in this study, and the average age was 20.67 ± 1.09. There were two main steps in the experimental process. In the first step, we examined the subjects’ refraction state and visual function, and then fitted disposable contact lenses with a corresponding refractive error. In the second step, the subjects gazed at an object from a viewing distance of 0.4 m and at a virtual image through a DMS, respectively, and the accommodative response and pupil size were measured using an open field autorefractor. Results: When the subjects gazed at the object from a distance of 0.4 m, or gazed at the virtual image through a DMS, the mean value of the accommodative response was 1.74 ± 0.43 or 0.16 ± 0.47 D, and the pupil size was 3.98 ± 0.06 mm or 4.18 ± 0.58 mm, respectively. With an increase in the viewing distance from 0.4 m to 2.285 m, the accommodative response and pupil size were significantly reduced about 1.58 D and enlarged about 0.2 mm, respectively. For three asterisk targets of different sizes (1 cm × 1 cm, 2 cm × 2 cm, and 3 cm × 3 cm), the mean accommodative response and pupil size through the DMS was 0.19 ± 0.16, 0.27 ± 0.24, 0.26 ± 0.19 D; and 4.20 ± 1.02, 3.94 ± 0.73, 4.21 ± 0.57 mm, respectively. The changes of the accommodative response and pupil size were not significant with the size of the targets (*p* > 0.05). In the low or high myopia group, the accommodative response of 0.4 m and 2.285 m was 1.68 ± 0.42 D and 0.21 ± 0.48 D; and 1.88 ± 0.25 D and 0.05 ± 0.40 D, respectively. The accommodative response was significantly reduced by 1.47 D and 1.83 D for these two groups. The accommodative microfluctuations (AMFs) were stable when a DMS was used; on the contrary, the AMFs were unstable at a viewing distance of 0.4 m. Conclusions: In this study, the imaging through a DMS extended the viewing distance and enlarged the image, and resulted in a reduction in the accommodative response and an increase in the pupil size. For the low myopia group and the high myopia group, the accommodative response and pupil size were statistically significantly different before and after the use of the DMS. The reduction of the accommodative response could be applied for the improvement of asthenopia.

## 1. Introduction

With the advancements in science and technology, people have changed their habits of receiving information and acquiring knowledge unprecedentedly. The frequent use of electronic digital products for reading books, newspapers and magazines has led to an increase in the frequency and time of near work [1]. Demir. P. et al. presented a study on the risk factors of myopia progression in Swedish schoolchildren in 2021. The results showed that 128 schoolchildren aged 12 ± 2.4 years old had an average of 5.3 ± 3.1 h of daily close activity and an average of 2.6 ± 2.2 h of daily outdoor activities, and it was found that time used for close activities was approximately double the time spent engaging in outdoor activities [2]. Research on near work has been a hot topic from the past to the present [3]. Besides Sweden, many countries have also reported relevant research in comparing near work and outdoor activities. The results of these studies indicated that the time spent in near work is more than that in outdoor activities, and most of the main studies ranged in age from 6 to 20 years old. Near work is often discussed with myopia progression [4,5,6,7,8,9].

It can be found that the time used for near work continues to increase. Although the amount of outdoor activity time has also increased, it is still far less than that used for near work. The influence of factors such as genes, race, region, and social culture also affect eye-use habits, which showed that with the maturity of civilized society and the advancement of science and technology. The phenomenon that near work increases during people’s school years is observed alongside many other changes that occur throughout that life stage. Williams, R. et.al. published a study objectively assessing near work in 2019. During the experiment, adult subjects were asked to wear a sensor device for detecting distance on weekdays and weekends. The results revealed the eye use at a distance of 0.6 m between object and eye was the longest, followed by 0.4–0.5 m [10,11,12,13,14,15,16,17,18].

Accommodation lag is a key factor related to myopia. Near work leads to a greater accommodation lag which may further the progression of myopia. Accommodative response to a near stimulus is an important component of clinical optometric assessments [19]. Many patients reported some symptoms during near work, some of which may be related to an inappropriate accommodative response. The accommodative response is generally less than the accommodative stimulus (the so-called accommodation lag) when viewing a close target (i.e., for stimulus levels that were greater than approximately 1D). An accommodation lag that exceeded the eye’s depth-of-focus resulted in blurred vision. However, the majority of patients were able to see near targets comfortably and clearly when their accommodation lag was less than the depth-of-focus of the eye.

The amount of accommodation lag is not constant for everyone, nor is the refractive diopter of the accommodating eye. Even when viewing a static object, the refraction fluctuates dynamically around the mean accommodative response within a limited range of 0.5 diopters (D). These small, rapid changes are called accommodation microfluctuations (AMFs), the range of which also varies from individual to individual. The accommodative microfluctuations (AMFs) can be measured in time and frequency domains. Many intrinsic and extrinsic factors influence the visual pathways that impact the magnitude of accommodative microfluctuations. Leahy et.al. reported almost no accommodative response when staring at infinity (far point), and that the fluctuations were closely related to cycloplegia. When the viewing distance was 0.66 m, the AMFs began to change and became unstable at 0.4 m. The decrease in pupil size and increase in the accommodative response resulted in an increase in accommodative microfluctuations [20]. Meanwhile, extending the viewing distance can decrease the accommodative microfluctuation [21,22]. The optical effect of the accommodative microfluctuations (AMFs) on the retinal image is to stabilize the image quality. Thus, both accommodation lag and AMFs decrease the optical quality of the retinal image. Accommodative relaxation can improve visual function and reduce eye fatigue [23,24,25,26]. In the present study, we proposed a double mirror system (DMS) to extend the viewing distance from 0.4 m to 2.285 m, so that the accommodative response could be relaxed [27,28]. In this study, the effects of extending the viewing distance on the accommodative response and pupil size of myopic adults using a DMS were discussed. The dependence of the accommodative response and pupil size on target size, in both the low and high myopic groups, was discussed. The accommodative response microfluctuations (AMFs) and pupil size variations were also presented. 

## 2. Methods

### 2.1. Design of the Double-Mirror System (DMS)

Figure 1 depicts the double-mirror system (DMS) as a method for extending the viewing distance. The system is composed of a concave mirror and a convex mirror. The convex mirror reduced the size of the image and enlarged the field of view while the concave mirror enlarged the image. Finally, the image was observed by the human eye. The diopters were +2.83 D for the concave mirror and −2.83 D for the convex mirror. The distance between the human eye and the concave mirror was 400 mm, 145 mm between the concave mirror and the convex mirror, and 280 mm between the convex mirror and the object (asterisk 3 cm × 3 cm). Based on the simulation, the distance between the eye and the virtual image was about 2.285 m, and the image magnification was 3.386×. Magnification is determined by image-size variation of the retina. When the viewer with an inter-pupil distance of 65 mm observes a target through the DMS, both eyes rotate to a vergence angle of 1.62° and observe an enlarged image at 2.285 m. The image, 0.254 mm wide and 0.195 mm high on the retina of both eyes, was analyzed using the commercial optical software, LightTools. A directly viewed image for the same target at 2.285 mm is 0.075 mm wide on the retina, which indicates that the magnification of the double system is around 3.386x in the horizontal direction.

### 2.2. Research Subjects

This study recruited 60 subjects between the ages of 18 and 22 years, with an average age of 20.67 ± 1.09 years. The inclusion criteria were: subjects without underlying eye or systemic diseases; those with spherical refraction spanning +1.0 D to −8.0 D; with astigmatism diopter ≥ −2.00 D; with a binocular visual acuity ≥ 0.1 logMAR; and with normal binocular vision.

Informed consent was obtained from all subjects, and the experiment was carried out according to the Declaration of Helsinki. Ethical approval was granted by the Institutional Review Board of Chung Shan Medical University Hospital in Taichung, Taiwan, ROC (Approval number: CS2-18104).

### 2.3. Research Process

The experimental procedure comprised of two steps. The first step was an assessment of the subjects’ primary visual function. Each subject received a basic examination of their refractive status, visual acuity (Snellen E Chart), phoria, stereoscopic vision and amplitude of accommodation. After the first examination, disposable contact lenses with a corresponding refractive error were provided to the subjects. With contact lenses on, we inspected again the subjects’ refractive state and visual acuity performance.

In the second step, we measured the subjects’ dynamic accommodative responses and pupil sizes. The viewing distances were assigned as follows: (1) the subjects viewed an object (asterisk 3 × 3 cm) at a distance of 0.4 m; then (2) a virtual image 2.285 m away through the DMS. An open-field autorefractor (Grand Seiko WAM WR-5500) was used to evaluate the dynamic accommodative responses and pupil sizes. Only data from the right eye were collected. During the examination, subjects were asked to blink naturally while fixing their gaze on the object or the virtual image. An open field autorefractor was used to perform a two-stage refraction examination on the subjects. First of all, the subjects’ refractive error at 6 m was measured. Secondly, the subjects’ refractive status at the given viewing distance (0.4 m or through a DMS) was assessed. The accommodative response was specified as the difference between the refractive status of the two-stage measurements and represented using the following formula:AR = RE − RS(1)
with AR being the accommodative response; RE the refractive error at 6 m; and RS the refractive status at the given viewing distance (0.4 m, or through a DMS). The units used for the accommodative response were diopters (D).

### 2.4. Data Analysis

The accommodative responses and pupil sizes were recorded, and the data were analyzed by SPSS Statistics 21.0 (IBM, New York, NY, USA). An independent sample *t*-test and paired samples *t*-test were used in this experiment for statistical analysis.

## 3. Results and Discussion

### 3.1. The Base Line of the 60 Subjects

The baseline of the 60 subjects is shown in Table 1. The refractive status for the male and female subjects was −3.25 ± 2.18 D and −3.68 ± 2.71 D, respectively. When the subjects gazed at an object from a viewing distance of 0.4 m, the mean value of the accommodative response for the male and female subjects was 1.66 ± 0.42 D and 1.88 ± 0.31 D, respectively. Among the subjects in this study, the females required a more accommodative response than the males when looking at close objects. The differences in age, spherical equivalent, accommodative response, and pupil size between men and women were compared. *p* value represented the probability of the test, and asterisk (*) is denoted statistically significant, *p* < 0.05.

When gazing from a distance of 0.4 m, the accommodative stimulus was 2.5 D. The mean value of the accommodative response for all subjects was 1.74 ± 0.43 D, which was a decrease of approximately 0.76 D. This phenomenon is known as “accommodation lag”. All subjects exhibited the phenomenon of accommodation lag at a near distance. Anderson et al. and Park et al. reported an accommodation lag of about 0.75 D and an accommodative stimulus of 3.0 D in 20-year-old subjects [29,30]. Corresponding to the results of this study, the accommodation lag was 0.76 D with an accommodative stimulus of 2.5 D, indicating the accommodation lags for both studies were similar.

### 3.2. Dynamic Accommodative Response and Pupil Size Distribution in Different Refractive States

Figure 2a,b show the accommodative response and pupil size distribution for the subjects according to their respective equivalent spherical diopters at a viewing distance of 0.4 m and 2.285 m, respectively. When the subjects gazed from a viewing distance of 0.4 m at a target located in front of a single eye, the accommodative response and pupil size were stable in each equivalent spherical diopter of the subjects. The mean of the accommodative response and pupil size was 1.74 ± 0.43 D and 3.97 ± 0.64 mm, respectively. When the refractive states of the subjects were completely corrected, the accommodative response was not significantly dependent on the equivalent spherical diopters of the subjects. In addition, the viewing distance was the key factor in the variation of the accommodative response. When the viewing distance was extended to 2.285 m through the DMS, the mean of the accommodative response and pupil size was 0.16 ± 0.47 D and 4.18 ± 0.05 mm, respectively. Comparing the subjects gazing through a DMS, the accommodative response was relatively larger and the pupil size is relatively smaller at the viewing distance of 0.4 m. Besides, with the increase in the equivalent spherical diopters, the accommodative response tended to decrease quite evidently, but the pupil size was relatively stable. The accommodation lag of the high diopter subjects was relatively larger through the DMS. The accommodation response and pupil size are affected by changes in the viewing distance. Additionally, accommodation response decreased while pupil size enlarged with distance, respectively. These results were expected and consistent with previous studies [31,32].

### 3.3. Accommodative Response and Pupil Size under Three Different Target Sizes

When the object was imaged through a DMS, the image was extended to 2.285 m and magnified 3.386 times. Therefore, we designed three different target sizes to understand the correlation between the target size and the accommodative response. The sizes of asterisk targets A, B, and C were 1 cm × 1 cm, 2 cm × 2 cm, and 3 cm × 3 cm, respectively. The means of the accommodative response for targets A, B, and C were 0.24 ± 0.16, 0.27 ± 0.24, and 0.26 ± 0.19 D, respectively, as shown in Figure 3a. The means of the pupil sizes were 4.20 ± 1.02, 3.94 ± 0.73, and 4.21 ± 0.57 mm, respectively, as shown in Figure 3b. The accommodative response and pupil size were not significantly different under the three different target sizes (*p* > 0.05). We speculated that because the viewing distance was far enough and the accommodative demand was smaller, the target size did not affect the change in accommodative response.

### 3.4. Accommodative Response and Pupil Size for the Low and High Myopia Groups

Table 2 and Figure 4 present the accommodative response and pupil size for the low and high myopia groups. First, the subjects’ refractive status was converted into an equivalent spherical diopter and separated into two groups. When the equivalent spherical diopter was larger than −5.00 D, the subjects were classified as belonging to the low myopia group, while those with less than −5.00 D were classified as belonging to the high myopia group. There were 44 subjects in the low myopia group and 16 subjects in the high myopia group. The means of the equivalent spherical diopter for the low and high myopia groups was −2.08 ± 1.71 D and −6.64 ± 0.91 D, respectively. In the low myopia group, the mean accommodative responses were 1.68 ± 0.42 D and 0.21 ± 0.48 D for the viewing distance of 0.4 m and 2.285 m, respectively. The accommodative response decreased by approximately 1.47 D, and showed a significant difference (*p* < 0.001) when extending the viewing distance through the DMS; in addition, the mean pupil sizes were 4.08 ± 0.54 mm and 4.27 ± 0.48 mm for the viewing distances of 0.4 m and 2.285 m, respectively. The pupil size also showed a significant increase (*p* < 0.001) when extending the viewing distance. In the high myopia group, the mean accommodative responses were 1.88 ± 0.25 D and 0.05 ± 0.40 D for the viewing distances of 0.4 m and 2.285 m, respectively. The accommodative response decreased by approximately 1.83 D, and showed a significant difference (*p* < 0.001) when extending the viewing distance through the DMS; in addition, the mean pupil sizes were 3.77 ± 0.87 mm and 3.97 ± 0.74 mm for the viewing distances of 0.4 m and 2.285 m, respectively. The pupil size also had a significant increase (*p* < 0.001) when extending the viewing distance. The result indicated that the accommodative response and pupil size were statistically significantly different before and after the use of the DMS for both the low and high myopia groups, as shown in Figure 4.

### 3.5. Accommodative Response Microfluctuations (AMFs) and Pupil Size Variation for the Low and High Myopia Subjects

To further understand accommodative response microfluctuations (AMFs) and pupil-size variations at a viewing distance of 0.4 m and through a DMS, this study randomly selected one low and one high myopia subject to conduct 20 s time frame analyses, as shown in Figure 5. The results revealed the mean accommodative response values to be 1.51 ± 0.27 D and 1.92 ± 0.29 D at the viewing distance of 0.4 m, and 0.19 ± 0.17 D and 0.23 ± 0.12 D through a DMS, for the low and high myopia subject, respectively. The results also showed stable accommodative microfluctuations (AMFs) using a DMS in contrast to unstable AMFs at a viewing distance of 0.4 m, as shown in Figure 5a,b. The mean pupil size values were 3.94 ± 0.31 mm and 3.88 ± 0.27 mm at the viewing distance of 0.4 m and 4.33 ± 0.18 mm and 4.38 ± 0.28 mm through a DMS, for the low and high myopia subject, respectively, as shown in Figure 5c,d. Using a DMS, the pupil size was enlarged by 0.39 mm and 0.50 mm, for low and high subjects, respectively. These results indicated that the relaxation of accommodation and increase in pupil size for both low and high myopic subjects can be achieved using a DMS.

## 4. Conclusions

This study discussed the accommodative response and pupil size of myopic adults using a DMS. The accommodative response was 1.74 ± 0.43 and 0.16 ± 0.47 D, and the pupil size was 3.98 ± 0.06 mm and 4.18 ± 0.58 mm, at a viewing distance of 0.4 m and through the DMS, respectively. With the increase in the viewing distance from 0.4 m to 2.285 m, the accommodative response and pupil size became significantly reduced by about 1.58 D and enlarged by about 0.2 mm, respectively. The result confirmed that the accommodative response decreased and the pupil size increased significantly through the DMS. The accommodative response and pupil size were not dependent on the target size (*p* > 0.05), because the viewing distance was sufficient and the accommodative demand was smaller, so the target size did not affect the change in the accommodative response. For both the low and high myopia groups, the result indicated that the accommodative response and pupil size were statistically significantly different before and after the use of the DMS. The accommodative response decreased significantly (*p* < 0.001) by approximately 1.83 D. In addition, the mean pupil size increased by 0.2 mm, with a DMS extending the subject’s viewing distance. The accommodative microfluctuations (AMFs) were stable when using a DMS; on the contrary, the AMFs were unstable at a near distance of 0.4 m. The accommodative relaxation and pupil size enlargement were achieved through a DMS. The reduction of the accommodative response may have the potential for application in improving asthenopia during near work.

## Figures and Tables

**Figure 1 ijerph-19-02942-f001:**
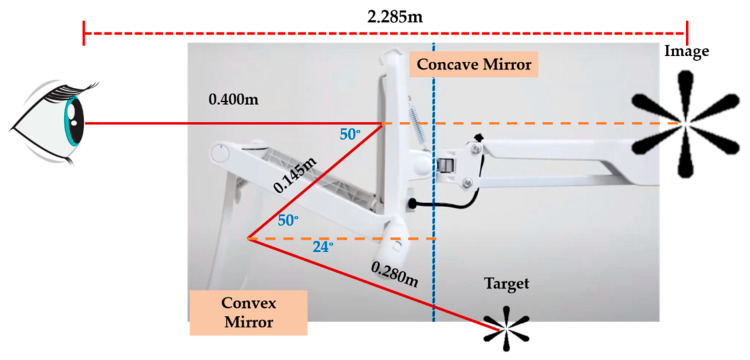
A double-mirror system (DMS) for the application of extending the viewing distance.

**Figure 2 ijerph-19-02942-f002:**
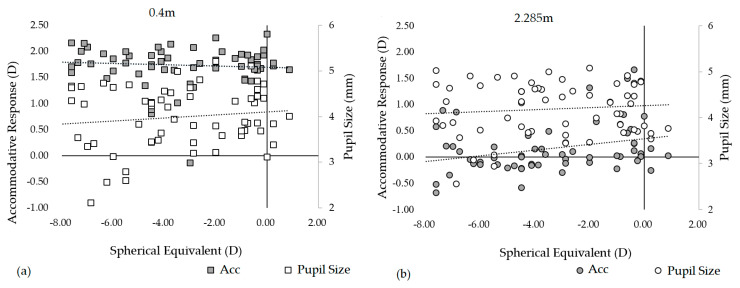
Accommodative response and pupil size distribution for the subjects at the viewing distance of (**a**) 0.4 m and (**b**) 2.285 m.

**Figure 3 ijerph-19-02942-f003:**
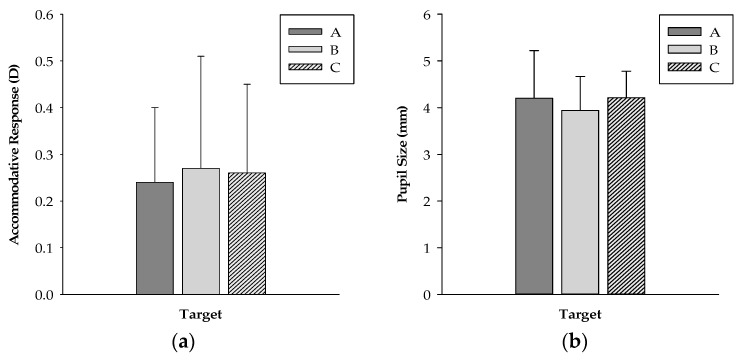
(**a**) Mean accommodative response; (**b**) Mean pupil size for targets A, B, and C.

**Figure 4 ijerph-19-02942-f004:**
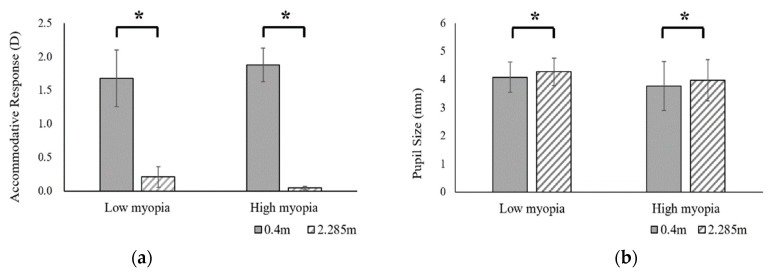
(**a**) Mean accommodative response; (**b**) Mean pupil sizes for the low and high myopia groups. (* *p* < 0.05).

**Figure 5 ijerph-19-02942-f005:**
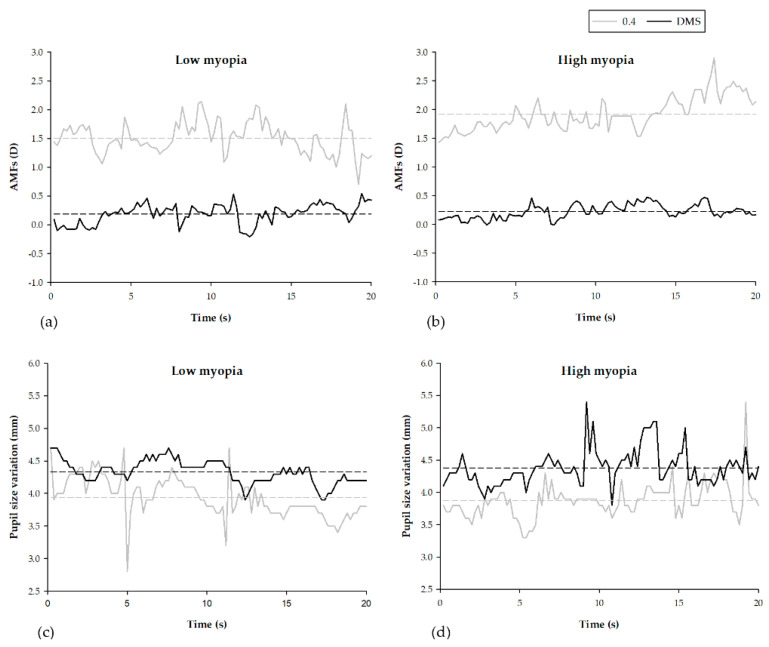
The accommodative response microfluctuations (AMFs) and pupil size variation at different viewing distances of 0.4 m, and through a DMS. (**a**,**b**) The accommodative response microfluctuations (AMFs). (**c**,**d**) The pupil size variation for the both of the low and high myopia subjects, respectively.

**Table 1 ijerph-19-02942-t001:** The base line of the 60 subjects.

	Base Line of Subjects			*p*
	Male (*n* = 30)	Female (*n* = 30)	All (*n* = 60)	
Age (y/o)	20.69 ± 0.97	20.72 ± 1.25	20.67 ± 1.09	0.882
Spherical Equivalent (D)	−3.25 ± 2.18	−3.68 ± 2.71	−3.45 ± 2.39	0.418
Accommodative Response (D)	1.66 ± 0.42	1.88 ± 0.31	1.74 ± 0.43	0.011 *
Pupil Size (mm)	4.09 ± 0.55	3.86 ± 0.71	3.97 ± 0.64	0.141

* *p* < 0.05.

**Table 2 ijerph-19-02942-t002:** Accommodative response and pupil size for the low and high myopia groups.

	Spherical Equivalent (D)		Viewing Distance		*p*
			0.4 m	2.285 m	
Low myopia (*n* = 44)	−2.08 ± 1.71	Accommodative Response (D)	1.68 ± 0.42	0.21 ± 0.48	<0.001 *
		Pupil Size (mm)	4.08 ± 0.54	4.27 ± 0.48	<0.001 *
High myopia (*n* = 16)	−6.64 ± 0.91	Accommodative Response (D)	1.88 ± 0.25	0.05 ± 0.40	<0.001 *
		Pupil Size (mm)	3.77 ± 0.87	3.97 ± 0.74	<0.001 *

** p* < 0.05

## Data Availability

The data that support the findings of this study are available from the corresponding author upon reasonable request.

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
