# Peer review of "Effects of Extended Viewing Distance on Accommodative Response and Pupil Size of Myopic Adults by Using a Double-Mirror System"

_ijerph, 2022, doi:10.3390/ijerph19052942_

Round 1
Reviewer 1 Report
The authors described an interesting work in which they examined the effect of extending the viewing distance on accommodative response. For this purpose, they used the double mirror system DMS. Below are my comments:
- In a description of the DMS there is information that the image magnification is 3.386x. It is not clear what the authors mean: is this an angular magnification of the eye image for the object viewed from 40 cm vs the object viewed by the DMS? If so, on what basis / why was such magnification applied? Or maybe it is a linear magnification for the image given by the DMS vs the size of the object positioned 28 cm in front of the first mirror of DMS? If so, in order for the eye to see the object through the DMS from a distance of 2.285 m at the same angle as the object directly from a distance of 40 cm, the magnification should be 5.7125x. With the linear magnification used in the examination, the eye would see the object through the system at a smaller angle - reduced in size.
Thus, which magnification is described, and why was it 3.386x? If the DMS were to be used in practice, it seems a good idea to keep the magnification for the eye not less than 1x in order to keep information about the details of the observed object, which may be important for people using this device.
- Lines 162-163 "Among the subjects in this study, the females needed more accommodative response than the males when looking at close objects." - Do the authors have any idea why the results differed for men and women?
- Table 1: What does the parameter p mean in the table? This information should be completed. If it is the p-value from t-test, what was compared?
- Figure 3 - The vertical axes have no markers (it is not known where, for example, 0.1 D is). Visually, the data does not correspond to the content of the article. No standard deviation marked (error bars).
- Asterisks in tables and figures - what do they mean? – lack of description.
- The Introduction contains difficult to understand, non-grammatical or not finished sentences (eg lines 69,70, 76).
Author Response
Dear reviewer,
Thank you for the precious suggestions on the manuscript, “Effects of Extended Viewing Distance on Accommodative Response and Pupil Size of Myopic Adults by Using a Double-Mirror System,” and we have made substantial revision to the manuscript based on the comments, as the details below.
- In a description of the DMS there is information that the image magnification is 3.386x. It is not clear what the authors mean: is this an angular magnification of the eye image for the object viewed from 40 cm vs the object viewed by the DMS? If so, on what basis / why was such magnification applied? Or maybe it is a linear magnification for the image given by the DMS vs the size of the object positioned 28 cm in front of the first mirror of DMS? If so, in order for the eye to see the object through the DMS from a distance of 2.285 m at the same angle as the object directly from a distance of 40 cm, the magnification should be 5.7125x. With the linear magnification used in the examination, the eye would see the object through the system at a smaller angle - reduced in size. Thus, which magnification is described, and why was it 3.386x? If the DMS were to be used in practice, it seems a good idea to keep the magnification for the eye not less than 1x in order to keep information about the details of the observed object, which may be important for people using this device.
Response:
Magnification is determined by image size variation on the retina. When the viewer with an inter-pupil distance of 65 mm observes a target through the DMS, both eyes rotate to a vergence angle of 1.62° and observe an enlarged image at 2.285 m. The image, 0.254 mm wide and 0.195 mm high on the retina of both eyes, was analyzed by the commercial optical software, LightTools. A directly-viewed image for the same target at 2.285mm is 0.075 mm wide on the retina, which indicates that the magnification of the double system is around 3.386x in the horizontal direction. (lines 125-131)
- Lines 162-163 "Among the subjects in this study, the females needed more accommodative response than the males when looking at close objects." - Do the authors have any idea why the results differed for men and women?
Response:
Based on the results of this experiment, we found that females needed an accommodative response of ~0.2 D more than males when looking at close objects. We need a larger sample size to observe whether there is still such a difference before making further inferences.
- Table 1: What does the parameter p mean in the table? This information should be completed. If it is the p-value from t-test, what was compared?
Response: An independent sample t-test and paired samples t-test were used in this experiment for statistical analysis (lines 169-170).
The differences in age, spherical equivalent, accommodative response, and pupil size between men and women were compared. P value represented the probability of the test, and asterisk (*) is denoted statistically significant, P<0.05. (lines 178-181)
Table 1. The base line of the 60 subjects.
|
|
Base line of subjects |
p |
||
|
|
Male (n=30) |
Female (n=30) |
All (n=60) |
|
|
Age (y/o) |
20.69±0.97 |
20.72±1.25 |
20.67±1.09 |
0.882 |
|
Spherical Equivalent (D) |
-3.25±2.18 |
-3.68±2.71 |
-3.45±2.39 |
0.418 |
|
Accommodative Response (D) |
1.66±0.42 |
1.88±0.31 |
1.74±0.43 |
0.011* |
|
Pupil Size (mm) |
4.09±0.55 |
3.86±0.71 |
3.97±0.64 |
0.141 |
*p < 0.05
- Figure 3 - The vertical axes have no markers (it is not known where, for example, 0.1 D is). Visually, the data does not correspond to the content of the article. No standard deviation marked (error bars).
Response: We have corrected the data 0.19± 0.16D to 0.24± 0.16D.(line221)
Scales for the vertical axes and error bars have been added in Fig.3.
- Asterisks in tables and figures - what do they mean? – lack of description.
Response: The description of asterisks (*) have been added in tables and figures.
- The Introduction contains difficult to understand, non-grammatical or not finished sentences (eg lines 69,70, 76).
Response: “The changes in people's lives are manifested in the school age, the phenomenon that the near work continues to increase in the stage. Williams, R. et.al. published a study on objectively assessing near work in 2019.” has been rewritten as “The phenomenon that near work increases during people’s school years is observed alongside many other changes that occur throughout that life stage. Williams, R. et.al. published a study on objectively assessing near work in 2019.” in lines 74-77.
kind regards,
With above responses and substantial revisions, please kindly consider its publication in Int. J. Environ. Res. Public Health. Thank you for your kind consideration and assistance.
Sincerely,
Shuan-Yu Huang
Professor,
School of Optometry,
Chung Shan Medical University,
syhuang@csmu.edu.tw

Reviewer 2 Report
Please see attachment

Author Response
Dear reviewer,
Thank you for the precious suggestions on the manuscript, “Effects of Extended Viewing Distance on Accommodative Response and Pupil Size of Myopic Adults by Using a Double-Mirror System,” and we have made substantial revision to the manuscript based on the comments, as the details below.
- In the first stage of the experimental process, information about cycloplegia is lacking when assessing the patient's refractive state.
Response: Regulations in Taiwan do not permit optometrists to use mydriatics. So instead, we utilized an open-field autorefractor to assess the patient's refraction in their accommodative relaxation state.
- In the article, in the part devoted to the examination of the patient, there is studies on accommodative response include more than one refractive measurements and use the mean result as the refractive value) and on this basis the value of the patient's refractive error is estimated and an adequate lens is placed in front of the patient's eye for the next stage of the examination. Unfortunately, there is a risk of undercorrection or hypercorrection of myopia in the studied patients, which may have an impact on the parameters tested in the second stage of the experiment.
Response: The refractive error assessments were executed carefully and contact lens were given to the subjects correspondingly. With contact lenses on, we inspected again the subjects’ refractive state and visual acuity performance. The risks of undercorrection or hypercorrection of myopia in the patients were extremely low extremely low. (lines 150-151)
- There is no information on how to assess visual acuity in the part concerning the examination of the patient. - there is information about the inclusion criterion for the study - the visual acuity of each eye individually not lower than 0.1 logMAR and normal (ie full visual acuity?) for the binocular visual acuity examination. Could the authors determine on which logMAR charts the visual acuity test was performed?
Response: In this study, the visual acuity test was performed using the Snellen E Chart. Normal binocular vision referred to the binocular vision function, not the visual acuity. The binocular vision examination included phoria, stereoscopic vision and amplitude of accommodation.(lines 148-149)
- In the research process section, the authors write about the use of adequate refractive error correction with a contact lens. Have the Authors used toric lenses in the presence of astigmatism? Did the Authors use a contact lens with a power equal to that obtained in the autorefractometer test?
Response:
Instead of toric lenses, we used spherical lenses based on the spherical equivalent we found through objective and subjective refraction examinations.
- The authors mention in the article additional tests in patients, such as the evaluation of stereopsia and phoria. There is no information on the results of these studies later in the text.
Response:
The stereoscopic and phoria evaluations were performed to ensure that the recruited subjects had normal binocular vision.
Kind regards,
With above responses and substantial revisions, please kindly consider its publication in Int. J. Environ. Res. Public Health. Thank you for your kind consideration and assistance.
Sincerely,
Shuan-Yu Huang
Professor,
School of Optometry,
Chung Shan Medical University,
syhuang@csmu.edu.tw

Reviewer 3 Report
The current work is an extension of the previous work done and published by this group. As compared to the previous publication, the sample size has been improved from 32 subjects to 60 subjects. In addition, myopic subjects have been categorized into high and low myopic. The current study thus improvises and validates the previous study. It may be interesting to see if DMS can be modified and applied on hyperopic subjects. Additionally, including subjects from different age groups may increase the merit of the work. If possible, it is suggested to either include different age groups or hyperopic subjects to extend the applicability of this work.
Author Response
Dear reviewer,
Thank you for the precious suggestions on the manuscript, “Effects of Extended Viewing Distance on Accommodative Response and Pupil Size of Myopic Adults by Using a Double-Mirror System,” and we have made substantial revision to the manuscript based on the comments, as the details below.
- The current work is an extension of the previous work done and published by this group. As compared to the previous publication, the sample size has been improved from 32 subjects to 60 subjects. In addition, myopic subjects have been categorized into high and low myopic. The current study thus improvises and validates the previous study. It may be interesting to see if DMS can be modified and applied on hyperopic subjects. Additionally, including subjects from different age groups may increase the merit of the work. If possible, it is suggested to either include different age groups or hyperopic subjects to extend the applicability of this work.
Response: Actually, a double-mirror system (DMS) can be applied to presbyopia subjects already. In the future, we can design and modify a DMS for hyperopic subjects and recruit hyperopic subjects from different age groups to extend the applicability of this work.
Kind regards
With above responses and substantial revisions, please kindly consider its publication in Int. J. Environ. Res. Public Health. Thank you for your kind consideration and assistance.
Sincerely,
Shuan-Yu Huang
Professor,
School of Optometry,
Chung Shan Medical University,
syhuang@csmu.edu.tw

Round 2
Reviewer 3 Report
The manuscript can be accepted in its present form.